# Psychological Impact of COVID-19 Pandemic and Related Variables: A Cross-Sectional Study in a Sample of Workers in a Spanish Tertiary Hospital

**DOI:** 10.3390/ijerph18073608

**Published:** 2021-03-31

**Authors:** Mónica Leira-Sanmartín, Agustín Madoz-Gúrpide, Enriqueta Ochoa-Mangado, Ángela Ibáñez

**Affiliations:** 1Department of Psychiatry, Ramón y Cajal University Hospital, 28034 Madrid, Spain; agustin.madoz@salud.madrid.org (A.M.-G.); enriqueta.ochoa@salud.madrid.org (E.O.-M.); angela.ibanez@salud.madrid.org (Á.I.); 2Ramón y Cajal Institute for Health Research (IRYCIS), 28034 Madrid, Spain; 3Department of Medicine and Medical Specialties, University of Alcalá, 28871 Madrid, Spain; 4Network Center for Biomedical Research in Mental Health (CIBERSAM), 28029 Madrid, Spain

**Keywords:** coronavirus disease 2019 (COVID-19), health personnel, psychological stress, risk factors, protective factors

## Abstract

*Introduction:* We intend to objectify the psychological impact of the COVID-19 pandemic on the workers of a tertiary hospital. *Methods:* All the workers were invited to an online survey. In total, 657 workers were recruited, including 536 healthcare workers (HCWs) and 121 non-healthcare workers (nHCWs). General Health Questionnaire-12 items (GHQ-12) was used as a screening tool. Sociodemographic data, working environmental conditions, and health behaviors were also analyzed. *Results:* inadequate sleep, poor nutritional and social interaction habits, misuse of psychotropics, female gender, COVID-19 clinical diagnosis, and losing a relative by COVID-19 were variables associated with higher probability of GHQ-12 positive screening. Significant differences between “frontline workers” and the rest were not found, nor was higher the probability of psychological distress in healthcare workers compared to non-healthcare workers. After 3 months from the peak of the pandemic, 63.6% of participants screening positive in GHQ-12 reported remaining “the same or worse.” *Limitations:* Causal inferences cannot be established. Retrieval and selection biases must be considered as the survey was not conducted during the peak of the outbreak. *Conclusions:* psychological impact of COVID-19 has been broad, heavy, and persistent in our institution. Proper assessment and treatment must be offered to all hospital workers.

## 1. Introduction

There is growing evidence about the fact that healthcare workers (HCWs), particularly those involved in direct assistance to infected patients (so called “frontline HCWs”), have been exposed to mental health issues while working during the SARS-CoV-2 pandemic in China [1,2,3,4] and all around the world [5,6,7,8,9,10], including Spain, the country where our study has been carried on [11,12,13]. The reasons for the psychological distress to which medical health workers were exposed might be related to the many difficulties of being safe at work, such as the initially insufficient knowledge about the SARS-CoV-2 virus, the lack of prevention and control strategies, the long-term workload for staff, the high risk of exposure to patients with COVID-19, the shortage of personal protective equipment (PPE), the lack of getting rest, and the exposure to critical life events (like infection and death of loved ones) [14,15].

Not only HCWs but also the general population has been psychologically affected by the COVID-19 pandemic [16,17,18], assuming then that fear and psychological distress are extended conditions in the general population subjected to an epidemiological context, such as that of the COVID-19 pandemic.

Experience gained in previous epidemic outbreaks (Severe Acute Respiratory Syndrome - SARS, Middle-East Respiratory Syndrome - MERS, Ebola crisis) shows that both HCWs and the general population can be affected by psychiatric symptoms of anxiety, depression, and posttraumatic stress disorder (PTSD) when exposed to these kinds of situations [19,20]. Less is known about the repercussion of the pandemics on non-healthcare workers (nHCWs) in hospitals, although some research carried in other countries show inconclusive results about whether they get more or less psychologically impacted than HCWs [15,21,22,23].

The COVID-19 outbreak has been an adaptative challenge for workers in the Spanish health system. In Spain, medical assistance is universal, free, and easily accessible, and offers wide coverage. Medical assistance to COVID-19 is being fully assumed by the Public Health System. During the first wave of the COVID-19 pandemic, large hospital infrastructures and primary health care had to modify their normal work routine, while also requiring an increase in the endowment of certain devices and human resources. In Spain, the main peak of COVID-19 contagion took place from the first weeks of March to the end of April 2020. In this period, all the healthcare institutions had to transform their infrastructure and human resources in order to assume the care demand raised by the COVID-19 outbreak [11]. Our Hospital, Ramón y Cajal University Hospital (Madrid), is a third-level hospital with a total number of 901 hospital beds before the COVID-19 outbreak. On March 30, the maximum number of hospitalizations for COVID-19 was reached in our institution with 1028 patients admitted for this cause from a total figure of 1293 admissions. The total number of COVID-19 admissions along the first wave of the pandemic outbreak was 2654, of which 2127 were discharged and 527 died. The number of ICU (Intensive Care Unit) and ventilatory support beds had to increase from 77 to 94 beds, reaching a total figure of 200 critical patients along the peak of the pandemic. Global mortality during that period was 567 patients, with 151 patients during the same period as the previous year.

Our hospital has a total number of workers of around 5250, and all the staff (including HCW and non-HCW) had to adapt their working routine to the emerging situation, which meant changes in their tasks and a huge increase in workload and shifts. During the first wave of the pandemic, the hospital had to hire a total of 640 new professionals, including 226 nurses, 201 nursing assistants, 47 physicians, 51 hospital porters, and 14 senior graduates. Of the entire workforce, a total of 1562 had to maintain some period of home isolation.

During the outbreak, a hotline for psychological support/psychiatric evaluation for professionals was implemented. In parallel, a program for psychological assistance for COVID-19 patients and relatives of patients was also developed. Individual assessment and treatment were implemented if needed.

In this work, we intended to study the scope of the emotional impact of the COVID-19 outbreak on our hospital workforce, and to determine whether HCWs were more intensely affected when compared to nHCWs, assuming the hypothesis that those performing clinical work in close contact with COVID-19 patients would be exposed to a higher risk of developing mental health issues. We also aimed to analyze a group of variables (demographic, professional, health-related, working environment-related), measuring their association with the presence of psychological burden on the workforce. The results obtained would facilitate the design and planning of preventive and therapeutic interventions to improve the mental health of hospital workers.

## 2. Materials and Methods

This is a cross-sectional study. We designed a form to make an online survey among all the staff working in Ramón y Cajal University Hospital during the COVID-19 outbreak. The survey was distributed online by institutional mailing, and it was uploaded to the Hospital’s intranet. All workers of different categories were encouraged to participate. In addition, 657 workers were recruited, which represents 12.4% of total workers, from which 121 were nHCWs and 536 were HCWs.

All participants were administered the *General Health Questionnaire* (GHQ), a validated tool for screening non-psychotic psychiatric disorders in the general population. The GHQ is a self-administered screening questionnaire, designed for use in consulting settings aimed at detecting individuals with a diagnosable psychiatric disorder [24]. In its original version, it had 60 items (GHQ-60). The 12-Item General Health Questionnaire (GHQ-12) [25] is the most extensively used screening instrument for common mental disorders, besides being a more general measure of psychiatric well-being. It is validated for its use in the Spanish population [26]. Individuals with positive screening were defined as those with a total score of GHQ-12 scale of 12 or beyond (using the Likert scoring system, ranging from 0 to 3 for each item assessed) [27]. It is assumed that those surveyed who scored 12 points or more reveal a situation of relevant emotional impact, which makes it advisable to rule out the presence of mental disorders.

The survey was conducted from 15 June 2020 to 25 July 2020 and was previously approved by the Ethics Committee for Clinical Research of the Hospital (study number 150/20, approved on 26 May 2020). Informed consent was required for all individuals before participating.

The form was divided in four sections, which grouped different kinds of variables: sociodemographic data (gender, age, type of familiar coexistence) and professional and health status during the pandemic (professional category, experience, type of activity, mental health personal history, infection by SARS-CoV-2, COVID-19 symptoms), stress factors of workers related to the working environment and activities during the pandemic, risk and protection behaviors outside the workplace during the pandemic, and the GHQ-12 scale.

Initially, a raw analysis of the results was carried out. To enhance the power of the analysis, variables were recoded and grouped the following criteria that were considered clinically relevant. Continuous variables were described by a mean and standard deviation (sd). Categorical variables were described by absolute and relative frequency. Inferential statistics of a student’s t-test was used for quantitative variables. The association between categorical variables was made using the Chi-square test. To study the association between psychopathological alterations and risk variables, a backward stepwise (Wald) logistic regression model was used, adjusting for those variables that were assumed, based on the bibliographic review, the raw results of our study, and the biological plausibility, which might have influenced the selected scale variable GHQ-12. The possible existence of interaction and confusion was explored. All contrasts were bilateral and with a significance level of p less than 0.05. All analyses were performed with the Statistical Package for the Social Sciences (SPSS), version 19 (IBM Corp. Released 2010. IBM SPSS Statistics for Windows, Version 19.0. Armon, NY, USA: IBM Corp.).

## 3. Results

Of 657 individuals which participated in our sample, 79.1% were women. The variable “age” was recorded in closed intervals of 10 years in width. The estimated average age was of 41.06 years (sd: 11.63). Furthermore, 84.2% of the sample exceeded the cut-off point (12 points or more total score, Likert scoring system) of the GHQ-12 test, suggesting the need to further explore the presence of any non-psychotic mental disorder. The average Goldberg score in our sample was 16.8 (sd: 5.5). In addition, 81.6% were healthcare workers (HCWs) while 15.3 were the average years of professional experience (sd: 10.9).

After analyzing the descriptive data of the sample, we first conducted an analysis to determine which variables were associated with positive screening in GHQ-12. A statistical significance was found for the following variables: female gender (*p* = 0.003), age (*p* = 0.016), professional category (being a nurse or a nursing assistant) (*p* = 0.001), having developed COVID-19 infection symptoms (*p* = 0.021), having been diagnosed of COVID-19 infection (*p* = 0.004), experiencing the loss of a relative/close person from COVID-19 (*p* = 0.044). Healthcare workers (HCWs) were not related to positive screening in a significant way (*p* = 0.268), but, because of the variable´s relevance, we decided to include it in logistic regression analysis. Frontline workers vs. second line workers did not fully reach statistical significance (*p* = 0.084), but because of the amount of evidence about a statistically significant association between frontline workers and psychological distress in previous research [1,28,29], we decided to include this variable in logistic regression analysis (Table 1).

As for health habits and risk behaviors (Table 2), the following variables were significantly more frequent in those individuals with positive screening GHQ-12 scores: inability to maintain adequate sleep hygiene habits (*p* < 0.001), inability to maintain adequate nutritional habits (*p* < 0.001), inability to maintain structured leisure activities and disconnecting from work (*p* < 0.001), inability to maintain adequate social interaction (*p* < 0.001), inability to maintain a regular physical activity routine (*p* = 0.008), and irregular use of psychotropic drugs (by self-prescription or prescribed by colleagues) (*p* < 0.001). Inability to regulate exposition to information in media and social networks nearly reached statistical significance (*p* = 0.076).

The results of the logistic regression analysis are shown in Table 3. We introduced the following variables in our analysis: inability to maintain adequate sleep hygiene habits, inability to maintain adequate nutritional habits, inability to maintain structured leisure activities and disconnecting from work, inability to maintain adequate social interaction, inability to regulate the exposition to media and social networks, inability to maintain a regular physical activity routine, increasing the use of alcohol or illicit drugs, use of self-prescribed (or prescribed by a colleague) psychotropic drugs, performing relaxation/meditation/mindfulness techniques, gender, age, type of cohabitation, experiencing COVID-19 symptoms, being diagnosed with a COVID-19 infection, having a risk for COVID-19, loss of a relative, professional category, and working in close contact with COVID-19 patients.

As for the evolution of psychological disturbances in our sample, it is striking that, after 2–3 months from the peak of the COVID-19 outbreak in Madrid, 59.9% of our sample responded that they felt emotionally “the same or worse” compared to then. If we looked exclusively at those who reached the cut-off point for GHQ-12 positive screening (therefore, those potentially with clinical disorders), the data were even more worrying: 63.6% were the same (34.9%) or worse (28.7%) than then.

Logistic regression was performed, meeting the basic assumptions of independence of errors, linearity in the logit for continuous variables, absence of multicollinearity, and lack of strongly influential outliers.

We introduced the following variables in step 1 of our analysis: inability to maintain adequate sleep hygiene habits, inability to maintain adequate nutritional habits, inability to maintain structured leisure activities and disconnecting from work, inability to maintain adequate social interaction, inability to regulate the exposition to media and social networks, inability to maintain a regular physical activity routine, increasing the use of alcohol or illicit drugs, use of self-prescribed (or prescribed by a colleague) psychotropic drugs, performing relaxation/meditation/mindfulness techniques, gender, age, type of cohabitation, experiencing COVID-19 symptoms, being diagnosed of COVID-19 infection, having a disease of risk for COVID-19, loss of a relative, history of mental disorder, professional category, and working in close contact with COVID-19 patients.

After exploring the association between the independent variables and the dependent variable using the backward stepwise (Wald) logistic regression model, predictive variables for GHQ-12 screening positive were those shown in Table 3. The risk of screening positive on the GHQ-12 scale among those who had inadequate sleep habits was 2.256 (95% CI: 1.325 to 3.842) greater than the same risk for those who had adequate sleep habits, if all the other variables remained constant. In addition, the factors of inadequate social interaction and inadequate nutritional habits multiplied the risk of exceeding the cut-off point in said scale by 3.169 (95% CI: 1.801 to 5.574) and 1.736 (95% CI: 0.933 to 3.229), respectively. The odds of exceeding the cut-off point on the GHQ-12 scale for females over the same odds for males was 1.736 (95% CI: 1.036 to 2.909). Using irregularly psychotropic drugs to control anxiety or insomnia multiplied the risk of exceeding the cut-off point on the GHQ-12 scale by 2.010 (95% CI: 1.047 to 3.860). Having been diagnosed of COVID-19 infection (2.024 (95% CI: 1.191 to 3.441)), and having suffered the loss of a relative or close friend due to COVID-19 (2.022 (95% CI: 1.047 to 3.904)) were factors related to a greater risk of screening positive on the selected scale.

## 4. Discussion

The COVID-19 pandemic has carried a great burden of psychological distress in the population of workers in our hospital, with no statistically significant differences between HCWs and nHCWs. Although this might be unexpected on a first approach, we must consider that the core stressful events about being in risk for infection by SARS-CoV-2 could be universal to all hospital workers, and they could be extensive to the general population in some sense. The results of a recent extensive review [17] comparing the levels of depression, anxiety, and other psychological symptoms reported in research carried worldwide conclude that patients are always the most affected subjects, being followed by HCWs and the general population (showing these two latter groups overlapping frequency figures, without significant differences).

In our institution, the COVID-19 outbreak challenged the hospital infrastructure and human resources to a great extent, with no clinical areas free from COVID-19-infected patients (except from Oncological and Trauma wards). Besides this, physicians from different specialties, with scarce training in treating this kind of pathology, were moved to treat patients admitted with COVID-19 pneumonia. Scarcity of individual protective equipment during the first wave of the pandemic in Madrid made professionals experience a great fear of getting infected and infecting significant others. The entire workforce, including HCWs and nHCWs, had to increase their workload and assumed a higher amount of pressure in the working environment, facing longer working shifts and limitation of rest days.

The results of our research showed that variables found associated with psychological distress in previous reports referred in the introduction section (as being frontline workers vs. second-line workers, or being HCW vs. nHCW) did not predict a higher risk of showing psychological distress by scoring positive in GHQ-12 in our sample. It is remarkable that keeping inadequate basic habits (sleep hygiene, nutrition habits, and social interaction) were variables that predicted a greater risk of scoring positive in GHQ-12 in our sample, so these behavioral variables were more significant than others (professional or working environment-related variables) in our results, contradicting our expectations. Promoting adequate hygiene-dietary habits and minimizing social isolation (respecting epidemiological restrictions) could be profitable to minimize the negative psychological burden of hospital workers under these circumstances.

A relevant potential limitation of the study is that the variable nHCWs groups together different types of professionals, with different levels of exposure to infected patients, ranging from those relatively preserved from contacts of infectious risk (i.e., administrative personnel, maintenance/technical staff) to those who keep close contact with patients affected by COVID-19 (i.e., hospital porters). Besides this, not all physicians or nurses were equally exposed to infected patients, since many of the assistance activities were differed or carried out through telematics. To limit that kind of bias in our study, variables were redefined in order to group our individuals in terms of direct exposure to COVID-19 infected patients or not. Despite this, in our sample, direct contact with patients infected with SARS-CoV-2 did not reach statistical significance in determining a higher probability of positive screening in GHQ-12.

Although nHCWs are not involved in clinical tasks with COVID-19-infected patients, the psychological impact they reported in our study could be determined by other variables, which were not explored in this paper. The association between these variables (e.g., growing workload, disruption of their usual working procedures, and extension of work shifts) and psychological distress in nHCWS could explain the absence of significant differences between HCWs and nHCWs risk to screen GHQ-12 positively in our study. This hypothesis needs further investigation to be confirmed. As some previous research points out, psychological impact on non-health workers could also be due to the lesser availability of adequate coping strategies and scarce knowledge of self-protection and prevention techniques in this group of workers [23].

In our research, the female gender appears to be a variable significantly related to positive screening in GHQ-12. This finding is consistent with the conclusions drawn in numerous previous papers [1,3,15,17,21,22]. There are also some references about women being biologically more disposed to develop higher levels of anxiety and PTSD than men [30].

In our research, the professional category of nurse did not reach statistical significance with regards to positive screening in GHQ-12 in logistic regression analysis, which is a common finding in most of the papers cited in our references. We must consider that the female gender is overrepresented among nurses/nursing assistant categories, which is a fact that can condition the interpretation of results. In a further exploratory analysis, when we performed linear regression analysis, we did find that nurses and nursing assistants tended to report significantly higher scores in GHQ-12 compared to other professional categories. Keeping adequate sleep and nutritional habits and maintaining adequate social interaction showed up as protective factors from developing positive screening in GHQ-12 during the pandemic in our sample. Losing a relative or a close friend by COVID-19 and being diagnosed of COVID-19 infection were variables associated with positive screening on an GHQ-12 scale.

A vast majority of the participants in our survey stated that they felt they were “the same or worse” than in the most severe moments of the COVID-19 outbreak in Spain in the months of March and April. This finding is consistent with previous work, which outlines that a psychological reaction to this kind of stressful situation may present with anxiety and fear in an initial phase, but may consolidate in persistent depressive and post-traumatic stress (PTS) symptoms in some individuals [31] Most of the research carried out during the initial phases of the COVID-19 outbreak outlined the need for preventive and screening strategies among health workers, but watching for the evolution of psychological distress over the long-term is also needed, to take care of the most chronically impaired individuals.

The limitations of our study include the fact that, being a cross-sectional research, associations between variables and positive GHQ-12 screening cannot be considered in terms of causality. Beside this, the fact that the survey was conducted several weeks after the critical stage of the pandemic may lead to a bias in recalling the psychological aspects experienced during the crisis. Furthermore, the time elapsed since the most acute moments of the sanitary outbreak could have led to the preferential participation of the most severely and chronically affected individuals. It would have been desirable that a larger proportion of the working staff had engaged in the survey, reaching 12.51% of the global number. The measurement tool used was a screening tool, validated in the general population, and, therefore, could underestimate certain dimensions of symptomatic discomfort in health sector professionals that might be specific to their care activities. As a screening tool, it does not allow the diagnosis of a specific disorder, but it points to a greater probability of mental disorders that should be evaluated in greater depth. Specific kinds of psychological disturbances linked to clinical assistance during the COVID-19 pandemic. Moral injury [32], vicarious traumatization [21], compassion fatigue [33], and burnout syndrome [34,35] have been assessed in previous research, but were not the targets of our work.

Previous research has suggested the need to implement a different kind of measure to prevent and minimize the psychological disturbance in healthcare workers, since we already know they are overexposed in many sources of stressful events during a pandemic [36,37,38,39].

Considering the results obtained, preventive and therapeutic strategies should be, perhaps, expanded to include non-health hospital workers, specifically targeting the groups identified as being at higher risk (women).

## 5. Conclusions

In our study, healthcare workers did not reach positive scoring for GHQ-12 test more frequently than non-healthcare workers, showing both categories a generalized, wide, and persistent psychological impact among their subjects during the COVID-19 pandemic. Moreover, those professionals placed in frontline duties did not score positively more frequently than others in a significant way. The female gender, health behaviors, and infection by SARS-CoV-2 of the individuals and/or their relatives were significantly associated with GHQ-12 positive screening as risk/protective factors.

These results make it advisable to further investigate the evolution of psychiatric symptoms in hospital workers, and to implement a proper preventive assessment and therapeutic programs to meet their needs.

## Figures and Tables

**Table 1 ijerph-18-03608-t001:** Bivariate analysis results: sociodemographic, health, and professional variables.

		GHQ-12 Positive Screening				Chi Square	Sig
		No		Yes			
		*n*	%	*n*	%		
Gender	Male	33	32.7%	103	18.9%	9.727	0.003
	Female	68	67.3%	442	81.1%		
Type of familiar coexistence	Alone or as a couple	61	59.8%	273	50.2%	3.183	0.084
	With dependents	41	40.2%	271	49.8%		
Healthcare worker	Yes	79	77.5%	448	82.2%	1.284	0.268
	No	23	22.5%	97	17.8%		
Working in close contact with COVID-19 patients	First line	46	45.1%	296	54.3%	2.927	0.105
	Second line	56	54.9%	249	45.7%		
Nurse/Nursing assistant/Healthcare Technician	Yes	33	32.4%	275	50.5%	11.292	0.001
	No	69	67.6%	270	49.5%		
Staff Physician and Resident	Yes	56	54.9%	372	68.3%	6.844	0.012
	No	46	45.1%	173	31.7%		
Presence of COVID-19 symptoms	No	71	69.6%	310	57.0%	5.657	0.021
	Yes	31	30.4%	234	43.0%		
Clinical diagnosis of COVID-19 infection	No	79	77.5%	337	62.5%	8.390	0.004
	Yes	23	22.5%	202	37.5%		
History of risk disease for COVID-19	No	92	90.2%	466	86.1%	1.233	0.339
	Yes	10	9.8%	75	13.9%		
Admission for COVID-19	No	101	99.0%	532	98.3%	0.262	0.512
	Yes	1	1.0%	9	1.7%		
Loss of a relative due to COVID-19	No	88	87.1%	422	77.9%	4.458	0.044
	Yes	13	12.9%	120	22.1%		
History of mental disorders	No	84	82.4%	457	84.3%	0.247	0.659
	Yes	18	17.6%	85	15.7%		
Current emotional state (compared to march-april)	The same or worse	44	43.1%	346	63.6%	15.038	<0.001
	Better	58	56.9%	198	36.4%		

*n* total for each variable may not match the sum of partial n given that some respondents did not correctly complete the GHQ-12.

**Table 2 ijerph-18-03608-t002:** Bivariate analysis results: protective habits and risk behaviors.

		GHQ-12 Positive Screening				Chi Square	Sig
		No		Yes			
		N	%	N	%		
Regular physical activity routine	No	50	49.0%	345	63.5%	7.623	0.008
	Yes	52	51.0%	198	36.5%		
Performing relaxation/meditation/mindfulness techniques	No	63	61.8%	342	62.9%	0.045	0.824
	Yes	39	38.2%	202	37.1%		
Sleep hygiene	Inadequate	26	25.5%	309	56.7%	33.513	0.001
	Adequate	76	74.5%	236	43.3%		
Nutritional habits	Inaquate	18	17.6%	247	45.5%	27.498	0.001
	Adequate	84	82.4%	296	54.5%		
Leisure activities	Inadequate	49	48.0%	394	72.6%	24.003	0.001
	Adequate	53	52.0%	149	27.4%		
Social interaction	Inadequate	20	19.6%	256	47.1%	26.451	0.001
	Adequate	82	80.4%	288	52.9%		
Exposure to media/social networks	Inadequate	31	30.4%	216	40.1%	3.444	0.076
	Adequate	71	69.6%	322	59.9%		
History of mental disorders	No	84	82.4%	457	84.3%	0.247	0.659
	Yes	18	17.6%	85	15.7%		
Irregular use of psychotropic drugs	No	89	87.3%	378	69.4%	13.705	0.001
	Yes	13	12.7%	167	30.6%		
Increasing alcohol intake/use of illicit drugs	No	85	83.3%	449	82.8%	0.015	1.000
	Yes	17	16.7%	93	17.2%		

**Table 3 ijerph-18-03608-t003:** Logistic regression analysis.

							95% CI for EXP (B)	
	B	E.T.	Wald	fd	Sig	Exp (B)	Inferior	Superior
Sleep	0.814	0.272	8.975	1	0.003	2.256	1.325	3.842
Nutrition	0.551	0.317	3.030	1	0.082	1.736	0.933	3.229
Social interaction	1.153	0.288	16.019	1	<0.001	3.169	1.801	5.574
Psychotropic drugs	0.698	0.333	4.396	1	0.036	2.010	1.047	3.860
Female gender	0.552	0.263	4.392	1	0.036	1.736	1.036	2.909
COVID-19 clinical diagnosis	0.705	0.271	6.787	1	0.009	2.024	1.191	3.441
Loss of a relative	0.704	0.336	4.400	1	0.036	2.022	1.047	3.904
Constant	−7.826	1.190	43.248	1	<0.001	0.000		

## Data Availability

Not applicable.

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
