# Peer review of "Psychological Impact of COVID-19 Pandemic and Related Variables: A Cross-Sectional Study in a Sample of Workers in a Spanish Tertiary Hospital"

_ijerph, 2021, doi:10.3390/ijerph18073608_

Round 1

Reviewer 1 Report

In this manuscript, the authors investigate the psychological impact of the COVID-19 pandemic on the workers of a tertiary hospital, more specifically, the manuscript provide an insight of the psychological impact of a specific Hospital in Madrid (Spain). The article is mostly well written and readable, and it provides significant content. Some major concerns arise however looking at the results section and in part at their discussion, which should be addressed by the authors.

The following list represents my major concerns about the manuscript:

As line numbering was missing in the document, the part in bold represents the part of the manuscript. This is then followed by an “-> “ and by my personal comment regarding that specific point.

 Introduction

  • Introduction structure:

-> The introduction seems to finish abruptly, figures regarding hospital’s workers are presented and the presence of psychological support and assistance programs are highlighted but no study rationale and hypothesis are present.  The introduction should follow a funnel structure, starting from a wider general area of interest and leading to the study rationale and hypothesis. This does not happen in this manuscript.

Materials and methods

  • The survey was conducted from June 15 to July 25, 2020 and was previously approved by the Ethics Committee for Clinical Research of the Hospital.

 -> is there an ethic number for this study?

  • Individuals with positive screening were defined as those with a total score of GHQ-12 scale of 12 or beyond (using the Likert scoring system, ranging 0 to 3 for each item assessed). It is assumed that those surveyed who scored 12 points or more reveal a situation of relevant emotional impact, which makes it advisable to rule out the presence of mental disorders.

 -> What is the choice of a total score of 12 or beyond based on? Please define

  • Qualitative variables, by absolute and relative frequency. Inferential statistics Student's t test was used for quantitative variables. The association between categorical variables was made using the Chi-square test or Fisher's exact test.

-> Qualitative variables and Categorical variables are synonyms, I suggest changing one of the two in order to be consistent within the paper.

Results

  • 1% were women. Estimated average age was of 41.06 years (sd: 11.63)

-> is this the mean of the sample? If yes, it is not correct to call it estimated, if not please clarify how it was estimated.

  • but because of the variable´s relevance we were further in our analysis later.

 -> this sentence is unclear, it seems something is missing. Please clarify.

  • Frontline workers vs second line workers did not fully reach statistical significance (p=0.084), but nearly did

->A p-value is the probability of finding a value as extreme (or more extreme) as the data you have observed, given the null hypothesis to be true. The statement in the manuscript is misleading. Being p= 0.05 a set cut off value to define NHST, its interpretation as “a trend” may be considered not correct. There is a lot of discussion in literature about this. I would suggest presenting the effect size of this analysis and discuss it. If you found a moderate or big effect size, it may lay the base to fully study the variable more in dept.  Effect sizes are more informative than p-values.

I suggest editing the table including also the effect size.

  • As for health habits and risk behaviors (table 2), the following variables were significantly more frequent in those individuals with positive screening GHQ-12 scores: inability to maintain adequate sleep hygiene habits (p=0.000); inability to maintain adequate nutritional habits (p=0.000);

-> Different p values are report as equal to zero. This is an error. When a software report p = 0.000 it doesn’t mean that p = 0. The software allows to inspect the value and verify the exact value of p.  If the software reports a number smaller than 3 decimal numbers p value should be reported as p < 0.001. This mistake is repeated across the rest of the paper and tables (see table 2 and 3), please amend it.

  • Logistic Regression.

-> It would be nice to see either in the text or in appendix if the assumptions of logistic regressions are met. Multicollinearity in particular.

-> Logistic regression write up should be improved and reported according to APA style. Predictor and criterion variables could be better and clearer defined. It is not clear from the description and from table 3 exactly which variables have been employed in the logistic regression.

  • As for the evolution of psychological disturbances in our sample, it is striking that, after 2-3 months from the peak of COVID-19 outbreak in Madrid, 59.9% of our sample responded that they felt emotionally "the same or worse" compared to then. If we looked exclusively at those who reached the cut-off point for GHQ-12 positive screening (there-fore those potentially with clinical disorders), the data were even more worrying: 63.6% were the same (34.9%) or worse (28.7%) than then

-> It is not clear why this is reported here instead of above where the frequencies are investigated. Reporting this after the logistic regression make it unclear and force the reader to scroll between pages and table.

Discussion

  • Besides fear of getting infected or fear of infecting others, nHCWs in hospitals could also have psychological distress because of other variables: growing workload, disruption of their usual working procedures and extension of work shifts. As some previous re-search points out, this greater impact on non-health workers could also be due to the lesser availability of adequate coping strategies and scarce knowledge of self-protection and prevention techniques in this group of workers (Tan et al., 2020).

-> Different variables are mentioned in this paragraph in the discussion, the results about these variables are however not discussed right away. This is quite confusionary.

  • One of this paper (G. Li et al., 2020) is a cross-sectional research focusing in women health workers in Wuhan, showing that professionals with more than 10 years of experience, having two or more children, and carrying a personal history of chronic diseases (including psychiatric disorders) were the ones which showed more acute stress
  • Li et al. (2020) paper is used as support of the results of the logistic regression. This is however not really related, while this manuscript investigates the two genders as predictor, Li et al. (2020) sample includes only female professionals. Consequently, the discussion of Li et al. (2020) even if relevant in other parts of the manuscript does not seem to be relevant in this specific paragraph.
  • Furthermore, these findings “about women being biologically more disposed to develop higher levels of anxiety and PTSD than men”, may suggest that gender may act as mediating variable between some of the IV and the DV employed in this study. This could be further explored.

  • In our research, the professional category of nurse did not reach statistical significance with regards to positive screening in GHQ-12 in logistic regression analysis
  • This statement seems to suggest that the variable professional category was employed in the logistic regression. This however seems to not be the case from table 3 and from the description of the regression model.

The following represents instead some minor concern such as typos or wording that may help improving the flow of the manuscript

Introduction

  • Including our country

->  I suggest rephrasing this. Our country refers to the nationality of the authors, despite the authors are from a Spanish institution and the paper is focused on a Spanish tertiary hospital, the link between that and the author nationality may be in some way not clear.

  • Apart from the emotional impact of the pandemic on the group of health workers, there are also recent bibliographic references focusing on the psychological impact expe…….

->  this part may be rephrased to improve the flow between paragraphs.

  • We must also look back and learn from the experience gained in previous epidemic outbreaks (SARS; MERS; Ebola crisis)……

-> I suggest rephrasing this sentence avoiding the use of the first person point of view.

  • On the one hand, large hospital infrastructures and primary health care had to modify their normal work routine

->  On the one hand is used to introduce a point of view, fact, or situation, followed by another that typically contrasts with it. The contrasting situation (on the other hand) is missing in the text.

Discussion

  • In a further exploratory analysis, when we per-formed linear regression analysis, we did find that nurses and nursing assistants tended to report significantly higher scores in GHQ-12 compared to other professional categories.

-> As these results are mentioned, would it be possible to provide these results as supplement material?

  • Another relevant issue is that nHCWs is actually a heterogeneous construct that groups together different types of professionals, with highly variable levels of exposure to infected patients, ranging from those relatively preserved from contacts of infectious risk (i.e. administrative personnel, maintenance/technical staff) to those who keep close contact with patients affected by COVID 19 (i.e. hospital porters). This could lead to an erroneous underestimation of the contact with SARS-CoV-2 infected patients in these workers. Besides this, not all physicians or nurses were equally exposed to infected patients, since many of the assistance activities were differed or carried out through telematics.

-> The subject of this paragraph is not clear. It seems initially that this paragraph is referring to limitation of the current study. However, the subsequent paragraph explains how researchers have overcome this issue. Please rephrase this paragraph in order to make it clearer for the reader.

Author Response

First, we would like to thank you for your exhaustive and kind review of our paper. It is full of interesting suggestions to improve our work. 

  • Introduction structure: corrections have been made to meet the "tunnel-structure" requirement.
  • Materials and methods: 
    • Number of Ethics Commitee for the research has been included
    • The choice of 12 or beyond as cutoff point in GHQ-12 has been based in a reference which is added in the manuscript. 
    • We have chosen "categorical" and used it through all the text to be consistent.
    • Average age was estimated, since the survey recorded the variable "age" asking the participants to select the correspondant 10-year interval which fitted their age (using closed intervals ranging from 20 to 70 years of age, in 10-years brackets). 
    • "Because of the variable´s relevance..." This sentece was confusing. We have corrected it, meaning that the variable´s significance in previous research made us consider to include it in logistic regression analysis.
    • Since the variables studied were dichotomous, the OR can be a good tool to estimate the effect size. In this case, comparing frontline workers with second-line workers in bivariate analysis with GHQ-12 scoring shows a small effect size: 0.067 (CI 95% 0.006-0.145). We decided to include the variable in logistic regression because it has been reported as significantly associated to psychological distress in previous research.
    • p=0.000 have been substituted for p<0.001
  • Logistic regression: 
    • Basic assumptions have been met, which has been reflected in the text
    • Logistic regression write up has been improved
  • Sentence about persistence of psychological distress has been moved to the descriptive statistics section
  • Discussion: all your sugestions about clearness and lack of coherence of some statements have been adressed and ammended in manuscript. 
  •  

With regards to linear regression, we are not intending to use these data in future research, but we include the figures related to our statement in this paper for your knowledge.

Coefficientsa

Model

Unstandardized coefficients

Standardized coefficients

t

Sig.

95% CI for EXP (B)

B

Std. Error

Beta

Inferior

Superior

Nurses and nursing assistants

-,884

,418

-,081

-2,114

,035

-1,706

-,063

a. Dependent Variable: Total scores in GHQ-12

  • Minor suggestions have been all taken into account, reviewed and ammended.

Kind regards, 

Mónica Leira

Reviewer 2 Report

Dear Editor,

COVID studies are considered an urgent need. COVID is affecting global health and the economy significantly. Therefore additional studies in COVID-related subfields are required. 
This study is focusing on the Psychological Impact of the COVID-19 pandemic and related variables. Therefore this study addresses important public health problems. 
The psychological impact of the COVID-19 pandemic on the workers of a tertiary hospital is analyzed.

I could not find the methodological error. Results seem interesting. 

The study is novel enough. 

Small suggestion:  Chi-square test or Fisher's exact test is used. Explanation of the reason for using these tests can be clarified more. The various test can be used for the statistical comparison of categorical data. The selection of the correct test is critical. It should be determined based on scientific criteria. Therefore authors can be clarified the reason for statistical test selections.

In my opinion, research is urgently needed to understand the different impacts of COVID. 

Decision: Minor Revision

Author Response

First of all, all the authors of this paper want to thank you for your kind review. 

With respect to your suggestion about the scientific criteria used in the statistic tests selection:  quantitative variables are studied using t-student tests. Categorical variables are studied using Chi-square test, or Fisher´s exact test (when the expected frequencies of the variable are lower than 5). In our study, Fisher´s exact test wasn´t finally used because these conditions did not appear in our sample. In order to avoid confusion, we are removing the reference to Fisher´s exact test in the manuscript

Kind regards, 

Mónica Leira

Reviewer 3 Report

The authors propose a study about the psychological impact of COVID-19 pandemi among workers in a Spanish tertiary hospital by providing health questionnaire-12 items (GHQ-12).

The proposed study is interesting but there are some points that the authors should better discuss.

The authors should be better described the novelties of their study with respect to existing ones. Furthermore, the authors should provide more details and discussion about the obtained results. The Discussion section also needs to be improved by analyzing the outcome of evaluation section.

I suggest to further analyze more recent approaches about the examined topics. In particular, I suggest the following papers to further investigate emotional state of the hospital workers and graph-based  approaches for supporting their analysis in the introduction section:

1) An emotional recommender system for music. IEEE Intelligent Systems.

2) An Epidemiological Neural network exploiting Dynamic Graph Structured Data applied to the COVID-19 outbreak. IEEE Transactions on Big Data.

Finally, I suggest to perform a linguistic revision.

Author Response

Thank you so much for your interesting and adequate suggestions.

We have proceeded to improve the discussion write-up, trying to highlight the novelties of our research with respect to previous studies. In our institution, COVID-19 outbreak challenged hospital infrastructure and human resources to a great extent, with no clinical areas free from COVID-19 infected patients (except from Oncological and Trauma wards). Besides this, physicians from different specialties, with scarce training in treating this kind of pathology, were moved to treat patients admitted with COVID-19 pneumonia. Scarcity of individual protective equipment during the first wave of the pandemic in Madrid made professionals experience a great fear of getting infected and infecting significant others. All the workforce, including HCWs and nHCWs, had to increase their workload and assumed a higher amount of pressure in working environment, facing longer working shifts and limitation of rest days.

 The results of our research showed that variables found associated to psychological distress in previous reports referred in the introduction section (as being frontline workers vs second-line workers, or being HCW vs nHCW) did not predict a higher risk of showing psychological distress by scoring positive in GHQ-12 in our sample. It is remarkable that keeping inadequate basic habits (sleep hygiene, nutrition habits, social interaction) were variables which predicted a greater risk of scoring positive in GHQ-12 in our sample, so these behavioral variables were more significant than others (professional or working environment related variables) in our results, contradicting our expectations. Promoting adequate hygiene-dietary habits and minimizing social isolation (respecting epidemiological restrictions) could be profitable to minimize the negative psychological burden of hospital workers under these circumstances.

We are also very thankful to your suggestion of including this new methodology for exploring emotional distress among healthworkers. We hope that our current work in implementing therapeutic and preventive strategies to help our workers to keep healthier may bring us the chance to use some of your methodological tools in further research.

Kind regards, 

Mónica Leira

Round 2

Reviewer 1 Report

The authors made a good effort in revising the manuscript and  all the major concerns have been addressed. The new version is clearer and has a better flow.

Issues related to statistics analysis have been addressed, in particular the p = 0.000. However some of them (which I guess have just been overlooked during revision) are still reported as p = 0.00.

I have listed them below:

Table 1 - Current emotional state(compared to march-april)

Table 3- Social interaction

Table 3 -Constant

Author Response

Dear Reviewer: 

We are very thankful to you for your kind and accurate suggestions, which have made our paper clearer and more rationally structured. We also thank your patience with regards to our missed minor corrections, which you have kindly pointed to us. We have just changed them in the way you asked, and they are now outlined in light blue colour. 

Regards,

Mónica Leira-Sanmartín

Reviewer 3 Report

I think that the authors have addressed all my concerns.

Author Response

Dear Reviewer:

We would like to thank your for you attention and interest in our work. 

Regards, 

Mónica Leira-Sanmartín